# In Vitro Modulation of Spontaneous Activity in Embryonic Cardiomyocytes Cultured on Poly(vinyl alcohol)/Bioglass Type 58S Electrospun Scaffolds

**DOI:** 10.3390/nano14040372

**Published:** 2024-02-17

**Authors:** Filiberto Rivera-Torres, Alfredo Maciel-Cerda, Gertrudis Hortensia González-Gómez, Alicia Falcón-Neri, Karla Gómez-Lizárraga, Héctor Tomás Esquivel-Posadas, Ricardo Vera-Graziano

**Affiliations:** 1Facultad de Química, Universidad Nacional Autónoma de México, Circuito Escolar de Ciudad Universitaria, Coyoacán, Ciudad de México 04510, Mexico; riveratf@unam.mx (F.R.-T.); esquivel.hector99@gmail.com (H.T.E.-P.); 2Instituto de Investigaciones en Materiales, Universidad Nacional Autónoma de México, Circuito Escolar de Ciudad Universitaria, Coyoacán, Ciudad de México 04510, Mexico; macielal@unam.mx; 3Facultad de Ciencias, Universidad Nacional Autónoma de México, Circuito Escolar de Ciudad Universitaria, Coyoacán, Ciudad de México 04510, Mexico; hortecgg@ciencias.unam.mx (G.H.G.-G.); maryaliciafalcon@ciencias.unam.mx (A.F.-N.); 4Cátedra CONAHCyT/Instituto de Investigaciones en Materiales, Universidad Nacional Autónoma de México, Circuito Escolar de Ciudad Universitaria, Coyoacán, Ciudad de México 04510, Mexico; karla.gomez@materiales.unam.mx

**Keywords:** PVA/bioglass hybrid, electrospun scaffolds, cardiomyocytes, physicochemical characterization, Ca^2+^ bioactivity, cell contractility patterns

## Abstract

Because of the physiological and cardiac changes associated with cardiovascular disease, tissue engineering can potentially restore the biological functions of cardiac tissue through the fabrication of scaffolds. In the present study, hybrid nanofiber scaffolds of poly (vinyl alcohol) (PVA) and bioglass type 58S (58SiO_2_-33CaO-9P_2_O_5_, Bg) were fabricated, and their effect on the spontaneous activity of chick embryonic cardiomyocytes in vitro was determined. PVA/Bg nanofibers were produced by electrospinning and stabilized by chemical crosslinking with glutaraldehyde. The electrospun scaffolds were analyzed to determine their chemical structure, morphology, and thermal transitions. The crosslinked scaffolds were more stable to degradation in water. A Bg concentration of 25% in the hybrid scaffolds improved thermal stability and decreased degradation in water after PVA crosslinking. Cardiomyocytes showed increased adhesion and contractility in cells seeded on hybrid scaffolds with higher Bg concentrations. In addition, the effect of Ca^2+^ ions released from the bioglass on the contraction patterns of cultured cardiomyocytes was investigated. The results suggest that the scaffolds with 25% Bg led to a uniform beating frequency that resulted in synchronous contraction patterns.

## 1. Introduction

In medicine, myocardial infarctions and other diseases of the cardiopulmonary system are considered high-risk diseases due to the high mortality rate in the world population [1]. In 2021, 20.5 million people died from cardiovascular diseases (CVDs): “High blood pressure, air pollution, tobacco use, and elevated LDL cholesterol were among the leading contributors to CVD deaths” [2]. In 2018, more than 202 million people worldwide suffered cardiac ischemia [3], so there is an urgent need to develop various strategies to prevent and treat them. Heart valves from metal alloys, as well as stents for the repair of defects in the coronary vessels, were developed [4]. Damage to cardiac tissue due to arterial lesions, infections, or myocardial infarctions is also being investigated and is expected to be repaired by tissue engineering [5]. Tissue engineering for the treatment of myocardial infarction involves the fabrication of three-dimensional scaffolds that promote cell proliferation and adhesion based on biocompatible natural or synthetic polymers. These scaffolds can be porous or dense patches. The human heart has a very low regeneration rate because the cell loss that occurs during infarction is replaced by fibrous avascular tissue that alters the heart rhythm. For these reasons, there is interest in developing biopolymers and bioglasses that regulate cardiac rhythm and regenerate the infarcted area of the myocardium [6]. It is also necessary to measure the effectiveness of the developed biomaterials on the activity patterns of the heart to restore it.

To develop biopolymers for cardiac muscle tissue repair, several polymers from natural sources were tested [7]. To mimic the mechanical and surface properties of the cardiac tissue to be regenerated, hybrid organic/inorganic scaffolds were developed using natural biopolymers; the most commonly used are gelatin [8], collagen [9], alginates [10,11], chitosan [12,13], and fibrin [13], whereas synthetic biopolymers such as poly(l-lactic acid) (PLLA) [14], polyurethane (PU) [15], poly(lactic-co-glycolic acid) (PLGA), poly (ε-caprolactone), (PCL), their respective copolymers, and, more recently, carbon nanofibers are less commonly used [16]. Poly (ethylene glycol) (PEG) alginate hydrogels have been used for the controlled release of drugs into cardiac tissue [17]. Hydrogels and PVA membranes have also been developed for the treatment of vascular emboli due to their biocompatibility and low protein adsorption [18]. The degree of hydrolysis of PVA and its molecular weight determine its mechanical properties, the degree of water solubility, viscosity, and chemical activity of the surface. The latter is due to the ability of the hydroxyl groups on the surface to interact with the surrounding medium via hydrogen bonds, which is mediated by the degree of hydrolysis of PVA. Based on these parameters, the properties of a biomaterial for cardiac tissue regeneration can be developed using a hybrid of PVA and a bioglass [19].

Bioglasses have been used since the second half of the last century as bioactive components in tissue engineering to stimulate angiogenesis. The most successful was Bioglass 45S5^®^, invented by Larry Hench in 1969, thanks to its good bioactivity and biocompatibility [20]. Bulk bioglasses are rigid, but in the form of microparticles mixed with ad hoc organic polymers, they enhance the mechanical properties of the polymer. Kargozar, S. et. al. [1] used the well-known bioglass 45S5^®^ [45SiO_2_-24.5CaO-24.5Na_2_O-6P_6_O_5_] mol%, obtained by casting, for cardiac applications. Still, since grinding produces particles with very different diameters, this leads to poor results for soft tissue engineering. Some bioglasses of this type can be obtained as nanoparticles by using the sol-gel method, such as 58S, whose composition is [58SiO_2_-36CaO-6P_2_O_5_] mol% [21]. In the present work, the synthesis, and properties of a type 58S bioglass (Bg) obtained by using the sol-gel method from tetraethyl orthosilicate (TEOS), triethyl phosphate (TEP), calcium chloride dihydrate (CaCl_2_∙2H_2_O), and deionized water were studied to investigate the effect of Ca^2+^ ions on cardiac functions [22].

Ca^2+^ ions in cardiac tissue affect functions that enable blood circulation in the body. It is known that the contractile work performed by these tissues is due to the tension and contractile efforts of myocytes and fibroblasts, which decrease with age, and to the contractile proteins surrounding cardiomyocytes. Muscle contractions are a function of intracellular Ca^2+^ levels in cardiomyocytes, and a decrease in these ions reduces myocardial conductivity and the development of cardiovascular disease [23,24]. One type of cardiovascular disease is ischemia, in which the loss of cardiomyocytes is later replaced by fibroblasts and scar tissue. These anatomical changes alter the electrophysiological properties of the myocardium and can lead to arrhythmias and heart failure [25]. The infarcted zone becomes asynchronous to the healthy tissue, and the heart expands and remodels to compensate, eventually leading to heart failure [26]. In this sense, the design of a tissue scaffold for the growth of cardiomyocytes in vitro must consider its conductive property, a property necessary for the expression of proteins and specific regulators that allow the reproduction of the vital functions of the tissue to be repaired [27,28]. Therefore, it is necessary to consider the release of Ca^2+^ ions from the scaffold to promote myofilament activation. Previous literature has not examined the effect of the amount of calcium provided by the bioglass nanoparticles on adhesion or the electrical activity of cardiomyocytes by using fluorescence patterns in electrospun hybrid scaffolds.

On the other hand, chick embryonic cardiomyocytes seeded on electrically conducting scaffolds show contractile activity and increased expression of several proteins involved in contractility and electrical intercellular coupling [29,30]. Therefore, tuning the electrical conductivity of scaffolds to achieve the desired scaffold properties and cell responses is of great interest. To date, electrospun scaffolds made of conventional polymers lack suitable mechanical and electrical properties. Therefore, the synchronous beating rate of cardiac cells cultured on these conventional materials has not been achieved [29,31].

Thus, this work aimed to estimate the contractile activity response of cardiomyocytes from chicken embryos on organic/inorganic electrospun hybrid scaffolds of poly(vinyl alcohol) with bioglass (58SiO_2_-33CaO-9P_2_O_5_) by stepwise addition of bioglass nanoparticles containing Ca^2+^ ions obtained by sol-gel.

## 2. Materials and Methods

### 2.1. Materials

Tetraethyl orthosilicate TEOS [Si(OC_2_H_5_)_4_]; Triethyl phosphate TEP [(C_2_H_5_)_3_PO_4_]; calcium chloride dihydrate (CaCl_2_∙2H_2_O); Hydrochloric acid (HCl); and PVA with an average molecular weight between 31,000 and 50,000 g/mol and degree of hydrolysis of 98–99%. Glutaraldehyde grade I, 25% in H_2_O, was used for the chemical crosslinking of PVA. All of the above chemical reagents were purchased from Sigma Aldrich (Burlington, MA, USA).

The following reagents were used for the culture of embryonic cardiomyocytes: Hanks’ saline solution composition (mM) 125 NaCl, 0.9 KCl, 3.6 NaHCO_3_, 0.3 Na_2_HPO_4_, 0.4 KH_2_PO_4_, 0.5 MgCl_2_, 0.4 MgSO_4_, 10 glucose, 2.9 sucrose, 9.9 HEPES, and 2.2 CaCl_2_, pH 7.2. Cell culture media M199 Gibco BRL 10468-018 with inhibitors of cell proteolytic activity were used to prevent cell damage. Fetal bovine serum (FBS), horse serum (HS), gentamicin (Gibco Life Tech., New York, NY, USA), and 17 mM HEPES (Sigma Aldrich) were added to enhance cell growth. A local supplier provided fertilized eggs according to USDA quality guidelines (Memo 800.65). Calcium Green^TM^-1 AM Cell Permeant, Invitrogen, Thermo Fisher Scientific (Waltham, MA, USA), was used as a fluorochrome, with emission at 506–531 nm in a Leica MZ75 fluorescence stereomicroscope, Leica Microsystems, Weztlar, Germany, equipped with a lamp with a 506–531 nm wavelength filter.

#### 2.1.1. 58S Bioglass Sol-Gel Synthesis

The synthesis of bioglass 58S was realized according to the report by Bui, X.V. et al. [22], with slight modifications. Bioglass (Bg) was synthesized in a closed vessel by a sol-gel reaction at room temperature using the following procedure: 100 mL of hydrochloric acid solution (HCl, 0.1 M) and 100 mL of TEOS were mixed in a reactor and stirred for 45 min. A volume of 23 mL TEP was poured into it with stirring. Finally, after 45 min, 36.68 g CaCl_2_∙, 2H_2_O was added with stirring for another 45 min to complete the reaction. The phases of polycondensation, solution gelation, and aging were carried out for ten days at room temperature in a closed container. Then, it was stabilized at 70 °C for three days and then at 120 °C for two more days. The product was carefully ground in an agate mortar to obtain nanometric particles. The obtained bioglass was characterized physicochemically.

#### 2.1.2. Electrospinning of PVA and Hybrid Scaffolds

As described elsewhere, a customized experimental electrospinning setup was used to fabricate the scaffold [32]. The conditions for electrospinning PVA and hybrid scaffolds were spinneret needle size and thickness of 0.7 × 30 mm 22 G (1/4), a spinneret-collector distance of 10 cm, a volumetric flow rate of 0.2 mL/h, an applied voltage of 15 kV, an electrospinning time of 5 h, and a flat collector area of 8 × 8 cm. Typical environmental conditions were 45% relative humidity and a temperature of 25 °C. PVA scaffolds collected on aluminum sheets were dried for 24 h at 70 °C in a vacuum oven to remove water residues.

In this study, the Bg composition was kept unchanged. However, to analyze the effect of calcium concentration on scaffold properties and cell response, scaffolds with different PVA/Bg ratios were prepared according to Table 1. To obtain scaffolds with a certain concentration of PVA/Bg, it was considered to keep the mass of PVA constant in all hybrid scaffolds and determine the amount of Bg that allows obtaining the concentrations indicated in Table 1. An aqueous solution of PVA and an amount of Bg were mixed at 70 °C for 1 h with continuous vigorous stirring and then poured into the electrospinning syringe. Table 1 also shows the concentration of CaO and Ca^2+^ based on the fact that each gram of Bg contains 0.099 g of calcium.

#### 2.1.3. Chemical Crosslinking of PVA/Bg Scaffolds

The PVA/Bg scaffolds were crosslinked as follows to prevent dissolution in water. In a separating funnel, 5 mL of HCl at pH = 2 and 500 µL of glutaraldehyde (GA) 25% in water were mixed; then, 25 mL of toluene was added. The mixture was stirred vigorously, and after the chemical reaction, the phases were separated, leaving the residue of glutaraldehyde dissolved in toluene in the upper part of the funnel. At the same time, the HCl migrated to the aqueous phase [33]. The scaffolds were cut with a round punch and immersed for 24 h in beakers containing 4.5 mL of the toluene-GA phase. Finally, the crosslinked samples were dried in a vacuum oven for 24 h at 70 °C.

### 2.2. Methods

#### 2.2.1. Characterization of Non-Crosslinked and Crosslinked PVA/Bg Hybrid Scaffolds

Scanning electron microscopy (SEM) images of crosslinked and non-crosslinked hybrid PVA/Bg scaffolds were acquired using a JEOL JSM-7600F, JEOL Ltd., Tokyo, Japan, field emission scanning electron microscope. A voltage of 10 kV and a current of 4.5 × 10^−11^ A were employed. The PVA/Bg scaffolds were sputter-coated with gold before being studied by SEM.

The diameters of the fibers were measured using the ImageJ software, Java 8. The obtained data were statistically analyzed using the program “R”, and finally, a histogram (Appendix A) was generated for each micrograph of the electrospun scaffolds.

In addition, to study the distribution of bioglass elements (silicon, calcium, and phosphorus) on the PVA matrix, a mapping of the elements that compose the bioglass was performed using JEOL-ARM-200F, JEOL Ltd., Tokyo, Japan, transmission electron microscopy (TEM) in the mode of STEM. A voltage of 200 kV and a current of 1.5 × 10^−11^ A were employed.

The nanofibers loaded with 20% bioglass were dispersed in ethanol with the help of ultrasound equipment (60 W/min), and a representative fiber of the tissue scaffold was coated with carbon and stacked on a 100-mesh copper grid (Cat # GD1010-Cu). Subsequently, the surface was analyzed by energy dispersive X-ray spectrometry (EDX) to determine the presence of the elements Si, Ca, and P.

The chemical structure of PVA and PVA/Bg hybrid scaffolds were analyzed in a wavelength range of 4000–400 cm^−1^ at a resolution of 4 cm^−1^ using a Fourier transform infrared spectroscopy with attenuated total reflectance (FTIR-ATR) spectrometer, Thermo Fisher Scientific Nicolet 6700, Waltham, MA, USA. Samples of scaffolds of 1 cm^2^ size were placed in the holder and adjusted with a screw.

Thermal transitions (glass transition temperature, Tg, and melting temperature, Tm) of the PVA/Bg hybrid scaffolds were determined by differential scanning calorimetry (DSC) using a TA Instruments DSC 2910 modulated DSC instrument, New Castle, DE, USA. Also, weight loss was determined by thermal gravimetric analysis (TGA) using a TA Instrument TGA 2950 high-resolution thermogravimetric analyzer, New Castle, DE, USA. The DSC analysis was performed under a nitrogen atmosphere in a heating/cooling/heating cycle. A first run from 20 °C up to 119 °C at 10 °C/min to remove residual solvent was followed by cooling to 20 °C at 10 °C/min and a second run from 20 °C up to 320 °C at 10 °C/min.

#### 2.2.2. Cardiomyocyte Culture on PVA/Bg Scaffolds

After physicochemical characterization, isolated primary embryonic chick cardiomyocytes were cultured on the PVA/Bg scaffolds to analyze their effects on cardiomyocyte survival, adhesion, and activity patterns. The pure PVA and hybrid scaffolds were cut into 0.25 cm^2^ pieces and sterilized under UV-C light for 25 min in 7.5 cm diameter Petri dishes into which dissociated cardiomyocytes were seeded. Primary cell culture was performed according to previously established procedures [34,35,36,37,38]. Ventricular cells collected from embryonic chick hearts in ovo at 7–8 days were enzymatically dissociated to obtain a density of 4 × 10^4^ cells/mL. An aliquot of approximately 20 µL was added to the scaffolds for 20 min to allow the cells to precipitate and adhere to the scaffolds. The remainder of the M199 medium supplemented with 5% FBS and 10% HS was added. Dishes containing cardiomyocyte-seeded scaffolds with media were incubated in a 5% CO_2_ atmosphere for up to 48 h at 37 °C. The cell media was replaced and replenished every 24 h under sterile conditions. Previous work has shown that the survivability of cardiomyocytes on different substrates is determined by their contractile status after a toxicity assay with a vital dye plus detergent [32,39]. Spontaneous cell contractility was observed at 36 °C ± 0.5 as the first sign of cell survival by ocular inspection on a microscope stage; this activity may persist for several days. Calcium Green-1 fluorophore was used to detect intracellular calcium fluctuations (ΔCa^2+^). The effect of calcium ion availability on the contractility of seeded cardiomyocytes was also investigated.

#### 2.2.3. Fluorescence Analysis

An aliquot of Calcium-Green-1 fluorophore (with a light absorption/emission frequency of 485/517 nm) was diluted in 10 µL of dimethyl sulfoxide (DMSO) and Pluronic acid, sonicated for 15 min, and made up to 500 µL with Hanks’ solution. The cell culture medium was carefully replaced with the fluorophore dilution and the dish was incubated at 37 °C for 25 min. Finally, the fluorophore solution was replaced again with Hanks’ solution and analyzed using a Leica MZ75 fluorescence stereomicroscope. Images were captured using a Leica DFC360FX, (Leica Microsystems, Wetzlar, Germany). All camera and exposure settings were made using the freely available Micro-Manager software 2.0, (National Institutes of Health). For offline analysis, one to four-minute recordings were made with an exposure time of 50–130 ms. Video images were analyzed by extracting fluorescence amplitude fluctuations in different regions of interest (ROI). All image and video sequences were analyzed using ImageJ 1.6.0_20 software (National Institutes of Health, Bethesda, MD, USA).

Cell images showing contractile activity were outlined with circles. The different scaffold regions were analyzed separately, and fluorescence intensity was plotted as a function of time. ROIs were selected using ImageJ Time Series Analyzer V3.2 tool based on the propagation patterns of the contraction. Finally, the generated data matrix was analyzed using OriginPro 9.0 software to obtain plots over time of contractile activity.

## 3. Results and Discussion

### 3.1. Characterization of Electrospun PVA/Bg Hybrid Scaffolds

#### 3.1.1. Nanofiber Diameter and Morphology by SEM

Fiber morphology and the effect of Bg concentration on fiber diameter were analyzed using SEM. Figure 1 shows the micrographs of the electrospun PVA/Bg scaffolds before and after chemical crosslinking. The morphology and diameter show a correlation with the Bg concentration as well as with the immersion of the hybrid scaffolds in the GA/toluene crosslinking solution. The average fiber diameter before crosslinking ranged from 0.127 to 0.338 µm, while after crosslinking ranged from 0.129 to 0.938 µm. Before PVA crosslinking, the fibers have similar fiber diameters and homogeneous surfaces at Bg concentrations below 15%, Figure 1a–c. However, at higher Bg concentrations, Figure 1d–f, the fibers do not have a cylindrical shape, with fibers containing 30% Bg having the largest diameter. The fibers have irregularities and a curved profile with Bg nanoparticles on their surface.

On the other hand, the increase in fiber diameter is also explained by the fact that glutaraldehyde and toluene molecules are introduced into the fibers, while some GA molecules participate in the chemical crosslinking reaction with PVA molecules. The absorbed toluene molecules cause swelling of the fibers and are interstitially located at the interface between the polymer and the bioglass particle.

The calculated uncertainties, given as direct measurements in Figure 2, include both statistical and nonstatistical uncertainties due to the resolution of the measurement system.

#### 3.1.2. Elemental Mapping of Bg Particles by TEM

The elements of interest in the Bg composition (Ca, Si, and P) were mapped by TEM to determine their spatial distribution within the fibers (Figure 2). It should be noted that the microscopic image of pristine PVA fiber was taken during a short exposure time to avoid the incident electron beam damaging the sample. In contrast, hybrids with a Bg concentration higher than 15% resisted the effect of the electron beam, which allowed a longer exposure time and better image resolution. The fibers with 20% Bg are a representative example. Figure 3a shows the nanofiber without highlighting the Bg elements. A regular surface can be seen without the formation of Bg agglomerates.

Figure 3b–d show a uniform distribution of the bioglass elements (silicon, calcium, and phosphorus). Figure 3e shows a uniform distribution of the superimposed elements in the nanofiber of the scaffold. As expected, the distribution of each element was homogeneous along the fiber segment. It is also observed that the degree of element coverage in the fiber is proportional to the molar percentage concentration, namely 58 mol% SiO_2_, 33 mol% CaO, and 9 mol% P. The analysis of TEM was performed on the scaffold whose average fiber diameter ranges from 196 to 369 nm, while the diameter of the particles is roughly estimated to be 20 nm or less.

#### 3.1.3. Functional Group Analysis by FTIR-ATR

The FTIR-ATR spectra of the PVA/Bg hybrid scaffolds with different Bg concentrations show the characteristic functional groups of PVA and Bg (Figure 4). Appendix A shows the wavenumber presented in the different scaffolds. Figure 5 illustrates the chemical crosslinking mechanism between the molecules of PVA and GA in an acidic medium, which was confirmed by FTIR-ATR analysis. According to this reaction mechanism, the OH- groups of PVA react mainly with the aldehyde groups of GA, forming O-C-O bonds associated with the vibration of the acetal bridges (900–1150 cm^−1^) and reducing the intensity of the OH- signal and therefore the hydrophilicity of the fibers [39,40,41]. PVA and Bg FTIR-ATR spectra are shown in Appendix A.

A decrease in the signal intensity of the PVA OH^−^ group at 3300 cm^−1^ was observed after PVA crosslinking, which may be attributed to the formation of new bonds by glutaraldehyde interaction (Appendix A). No signals were observed from GA or toluene residues used for chemical crosslinking. It should be noted that the signals corresponding to Si-O-Si bonds were not perceptible at concentrations below 20% Bg in the PVA matrix.

#### 3.1.4. Thermal Analysis by DSC and TGA

##### Differential Scanning Analysis, DSC

The T_g_ and T_m_ of the samples before and after PVA crosslinking are shown in Figure 6. The DSC thermogram of the non-crosslinked PVA/Bg samples showed an endothermic transition between 50 °C and 61 °C because of evaporation (T_Ev_, evaporation temperature) of low-molecular-weight residual solvents. After evaporation of the residual solvents and cooling, the second heating cycle clearly showed that the T_g_ was in the range between 63 °C and 88 °C [42].

The glass transition (T_g_) of PVA depends on the degree of hydrolysis. The T_g_ reported for PVA with a degree of hydrolysis below 98–99% is close to 85 °C [43,44,45,46]. The melting temperature, T_m_, of PVA also depends on the degree of hydrolysis. The reported values range from 180 °C to 230 °C. The PVA used here exhibited a T_m_ of 218 °C. It was reported that the crystalline phase of PVA has an average T_m_ of 230 °C. For partially hydrolyzed PVA, some authors have reported that PVA has an average T_m_ between 180 °C and 190 °C, [5,47], while other authors have reported a T_m_ of 209 °C [48].

The T_m_ may also vary depending on the average molecular weight of the PVA. The T_m_ of the PVA/Bg hybrid scaffolds shift slightly to higher temperatures as the Bg concentration increases (Table 2). Bg is assumed to adsorb energy before transferring into the polymer, which is enhanced by increasing the Bg concentration in PVA. This effect is more pronounced in hybrid scaffolds where PVA is chemically crosslinked with GA. For PVA/Bg samples, the T_m_ was observed between 218 °C and 300 °C, Appendix A.

##### Thermogravimetric Analysis, TGA

The TGA thermograms of PVA/Bg scaffolds before and after crosslinking showed three clearly defined stages for both groups of scaffolds. The weight loss resulted from the Bg content ranging from 5 to 30%, Table 3. For the crosslinked scaffolds, all samples showed a weight loss of 3 to 16% in the first step, which ranged from 40 °C to 200 °C, due to the presence of residual solvents such as ethanol and water adsorbed on the scaffold. For the non-crosslinked samples, the weight loss ranged from 3 to 10%. In the second step, at 200–350 °C, all scaffolds mostly lost weight due to thermal degradation of PVA molecules. PVA gradually loses OH^−^ groups at temperatures above 200 °C, which decreases its degree of hydrolysis. As the temperature continues to rise, the weight loss is attributed to degradation products of the hydroxyl groups on the PVA backbone such as acetic acid. For non-crosslinked PVA/Bg fibers, the weight loss ranged from 50 to 66%, while for crosslinked samples, the weight loss ranged from 28 to 66%, with Bg ranging from 5 to 30%. As expected, crosslinking increased the thermal stability of the hybrid composites. The third range, from 350 to 600 °C, showed the mass loss of gases released by the complete combustion of PVA [48,49], and, in general, the crosslinked scaffolds were more stable in the previous two steps, showing a higher weight loss in the third step than the non-crosslinked scaffolds.

The TGA thermograms of PVA/Bg hybrid scaffolds before and after crosslinking are shown in Appendix A, in addition to PVA and Bg thermograms, respectively (Appendix A).

#### 3.1.5. Biological Interaction of Ca^2+^ in Hybrid Scaffolds with Cardiomyocytes

Cardiomyocytes were seeded on scaffolds and cultured in a medium providing appropriate ionic and pH conditions and necessary nutrients, including O_2_ and CO_2_. The electrolytic medium conditions were enriched with Ca^2+^ and Si^2+^ ions released from the PVA/Bg scaffolds. The effect of calcium on the contraction patterns of cardiomyocytes on the hybrid scaffolds was analyzed by fluorescence microscopy. Figure 7 shows representative images of the behavior of cardiomyocytes on crosslinked and non-crosslinked scaffolds. The living cell population could be identified by its contractile activity, which is seen in the green fluorescence color. It is due to intracellular calcium fluctuations that appear to be associated with increased Bg concentration releasing Ca^2+^ and Si^2+^ ions. That is when the cells are depolarized to contract, implying that their molecular system is functional, including proteins like actin, myosin, troponin, tropomyosin, ATP, Ca^2+^, etc.

The images correspond to a layer of thousands of cells, also called a cell monolayer. Therefore, it is not possible to distinguish individual cells with a Leica microscope at 4× magnification, which was used.

Images of PVA/Bg scaffolds at Bg concentrations below 15% in Figure 7a*–c* show independent cell aggregates (niches) with heterogeneous contractile activity. The formation of independent niches hinders the establishment of a wavefront that could reach distant cells and prevents synchronous periodic activity. On the scaffold with 15% Bg, a region of cell niches was observed on a thin layer of cells, Figure 7d*. This cell layer established an incipient but weak connection between cell aggregates that did not promote the spread of contractile activity over the entire scaffold surface. In contrast, the scaffolds with 20%, 25%, and 30% Bg, Figure 7e*–g*, allowed the formation of a strongly connected cell layer that covered the entire scaffold surface, also suggesting high cell adhesion. In this case, the contractile activity of the cells was able to spread along the scaffold, favoring a homogeneous, synchronous, and periodic activity.

Analysis of cardiomyocyte contractile activity shows that regions of interest (ROIs) exhibit homogeneous, synchronous, and rhythmic activity. Remarkably, inactivity is not observed in any ROIs throughout the time interval of the experiment (50 s). Figure 8 shows the sequence of contraction pulses in embryonic cardiomyocytes applied to crosslinked scaffolds, including the type and intensity of cellular electric pulses. Again, at Bg concentrations below 20%, cells developed in independent niches and did not exhibit homogeneous contraction pulses. At Bg concentrations of 20%, 25%, and 30%, the cells formed a uniform cell layer in the ROIs and showed a homogeneous sequence of contraction pulsations. Overall, it was observed that the electrical activity patterns in cardiomyocytes seeded on the electrospun scaffolds strongly depend on the Bg concentration, especially on the availability of Ca^2+^ ions.

#### 3.1.6. Proposed Model of Ca^2+^ Ion Dissolution with Time for the Interaction with Cardiomyocytes

Next, a schematic showing the evolution of both the size of the radius of PVA fibers containing Bg and the concentration of Ca^2+^ ions near the fibers as a function of time when the scaffold is embedded in the body fluid is shown in Figure 9. As can be seen in the microscopic images of the Bg nanoparticles (Figure 2), the Ca^2+^ ions are uniformly distributed in the fibers, both on the surface and throughout their volume, as shown in the following schematic, Figure 9. Adenyika [50] showed that the dissolution of PVA in water is a function of concentration. At low concentrations (0.2% *w*/*v*), PVA particles dissolve in 30 to 70 min. Muriel [51] concluded that the rate of dissolution is a function of the pH of the medium: at a higher pH, dissolution is reduced to a few minutes. In studies performed with Bioglass 45S5 [52], it was found that the dissolution speed depends on the particle size. It was observed that particles from 2 µm to 16 µm dissolve in the first few seconds after being placed in a physiological medium, and after 20 min, the dissolution rate stabilizes. Based on these tests, the following qualitative model is proposed to explain the dissolution of Ca^2+^ ions in water as a function of time (Figure 9), assuming that all nanofibers decrease their radius uniformly with time.

Once the nanofibers come into contact with the aqueous environment, Si^2+^, Ca^2+^, and PO_4_^3−^ ions are released. In particular, the release of Ca^2+^ ions tends to increase the pH of the solution. In contrast, the pH of the aqueous medium decreases with the onset of PVA dissolution and balances the pH of the medium [8]. This process ends when the PVA is completely dissolved.

During the first phase (t_o_–t_1_), the dissolution of Ca^2+^ ions from Bg increases continuously from zero to a maximum value. Thereafter, the dissolution of Ca^2+^ ions decreases while the radius (r) of the fibers continues to increase by swelling to a maximum (t_2_). After t_2_, both Ca^2+^ and the fiber radius decrease at different rates until t_3_, where both rates become similar, followed by the complete dissolution of the fibers, Figure 9a.

In addition, there is water absorption and adsorption that occur in the fibers from t_o_ to t_1_. The fiber radius increases from an initial value of r_0_ to a higher value of r_1_ at t_1_ due to swelling (r_1_ > r_0_); here, the Bg concentration in the aqueous media reaches a maximum. After this point, the Ca^2+^ concentration continuously decreases to its minimum value; however, due to swelling, the fiber radius continues to increase to a maximum value (r_2_) (Figure 9b). Thereafter, due to erosion of the PVA and dissolution in the aqueous medium, the fiber radius decreases to r_3_ until the fibrillar structure is completely dissolved (t_5_), Figure 9b. It is worth mentioning that the swelling phenomenon of the fibers is a consequence of the dissolution process of a polymer. This is described by a series of steps, including the diffusion of the solvent in the polymer matrix and the disentanglement of the polymer chains by a swelling process [53]. In the case of crosslinked polymers, the chains cannot slide between them due to the network structure formed by covalent bonds, and, therefore, infiltration by the solvent begins, causing swelling of the polymer structure.

Fluorescence images of seeded cardiomyocytes (Figure 10) show that the cells remain on the scaffold surface. There is no cell migration into the fiber interior because the pore size of the fibrillar structure is smaller than that of the cardiomyocytes. The mechanism by which ionic migration of Ca^2+^ to the interior of the cytoplasmic membrane occurs is shown schematically in Figure 10. After contact with the scaffold, cardiomyocytes adhere to the hydrophilic nanofiber surface; this adhesion is further enhanced by the release of Si^2+^, Ca^2+^, and PO_4_^3−^ ions. When these ions dissolve in the culture medium, they interact with the cardiomyocytes across their cytoplasmic membrane.

The electrospun scaffolds act as a synthetic extracellular matrix (ECM) that stimulates cardiomyocytes to adhere and orient to the nanofibers, promoting the formation of epimysial and endomysial fibrils. These fibrils help cells form a coherent tissue that allows nutrient diffusion and gas exchange, including CO_2_ and O_2_ [54,55,56,57].

Some biopolymers loaded with metal ions generate electric current depending on the ion concentration [58]. As shown in Figure 8, Si^2+^, Ca^2+^, and PO_4_^3−^ ions enhance electrical conduction in the case of PVA/Bg scaffolds in comparison with PVA pristine scaffold. After cardiomyocytes are seeded on the scaffold, the culture medium promotes cell interconnection through endomysium and epimysium fibrils, which form active cellular networks that provide mechanical support. Perimysium fibrils also facilitate mechanical work during the contraction and expansion cycles of these cell assemblies through Ca^2+^ exchange. All of these fibrils support action potential function at the cellular level [55].

Depending on Bg concentration, Ca^2+^ ions diffuse into cardiomyocytes and interact with the surface of the cytoplasmic membrane. The membrane senses the presence of these ions as they alter the electrical potential between intracellular and extracellular space by flowing through selective L-type Ca^2+^ channels (Cav1.2), [59,60]. Ca^2+^ ions are crucial intracellular messengers for cellular functions. An increase in concentration in the periphery of the cytoplasmic membrane leads to various cellular responses, including activation of the Ca^2+^ ion channel, ion current or flux, enzymatic activity, neurotransmitter and hormone secretion, and muscle contraction. In other cell types such as stem cells, proliferation, differentiation, and apoptosis are also influenced by Ca^2+^. Once these channels are opened, Ca^2+^ flows into the cytosol and causes electrical excitation of the cell [58].

The discharge of intracellular deposits activates the influx of Ca^2+^ ions and is associated with membrane depolarization. There are two recognized Ca^2+^ ion channels: L-type and T-type. The first type is activated at a more negative potential difference, i.e., it operates with a low threshold, while the second type operates with a high threshold.

T-type Ca^2+^ channels are functionally expressed in embryonic hearts, whereas they are expressed at higher densities in adult cardiac tissue. They are found in pacemaker cells, so these channels play an important role in pacemaker synchrony. In most cases, these cells contribute to the automatic function of the heart [61].

Putney proposed that the emptying of cellular depots causes the entry of Ca^2+^ from the extracellular medium into the internal depots that form the sarcoplasmic reticulum and from these into the cytosol. During recovery, active pumps such as sarcoendoplasmic reticulum calcium ATPase (SERCA) fill the reservoirs with Ca^2+^ from the cytosol rather than from the extracellular environment. In the plasma membrane, Ca^2+^ channels allow Ca^2+^ from the extracellular environment (1 mM Ca^2+^) to enter the cell interior (100 nM Ca^2+^) in favor of a larger electrochemical gradient [54,62,63,64,65,66].

## 4. Conclusions

Electrospun scaffolds were successfully fabricated using hybrid blends containing PVA as an organic matrix with Bg nanoparticles in a typical composition of 58% SiO_2_, 33% CaO, and 9% P_2_O_5_. The sol-gel technique used here to synthesize Bg made it possible to obtain nanoparticles with diameters of 5–20 nm, which facilitated their uniform incorporation into fibers with diameters ranging from 130 nm to 340 nm. Chemical crosslinking of PVA and increasing the Bg concentration allowed the hybrid nanocomposite scaffolds to be thermally stabilized. The proposed model for the change in fiber radius and calcium concentration with time illustrates the process of nanofibers dissolution for better understanding when the scaffold is placed in a physiological medium.

Isolated embryonic ventricular cardiomyocytes seeded on the fibrous scaffolds were successfully examined in vitro for survival, surface adhesion, and spontaneous activity patterns. The second graphical model describes a mechanism by which Ca^2+^ diffuses from the ECM nanofiber interface to its entry through calcium ion channels into cardiomyocytes. Cell viability analyses demonstrated the influence of Ca^2+^ on the adhesion and contractility of cultured cardiomyocytes. Cardiomyocyte adhesion was enhanced by fiber morphology resembling an ECM and the availability of Ca^2+^ ions released by Bg. Calcium ions promote the modulation of spontaneous activity of cardiomyocytes in vitro and enhance the activation of myofilaments. If the cardiomyocytes do not adhere to the scaffold, they float in the saline solution and are easily washed out. The results show that the crosslinked PVA/Bg scaffolds at concentrations of 20%, 25%, and 30% provide the best physicochemical conditions for cardiomyocytes to spread evenly on the surface of the scaffolds and show contractile pulses of activity that spread homogeneously over the entire surface. These scaffolds have a better release of Ca^2+^, which gradually dissolves from the scaffolds into the culture medium for the cardiomyocytes, inducing the generation of uniform electrical pulses that lead to the observed synchronous contraction patterns in vitro (video).

At least from the results obtained in this work, it can be confirmed that the morphological and chemical stability of the electrospun fibers that make up the scaffolds is stable over a period of 48 h, during which the fluorescence study with the cardiomyocytes was carried out. However, the stability of the scaffolds should be determined at different time intervals.

The results obtained suggest that these hybrid scaffolds are promising candidates for studying the regeneration of infarcted myocardium.

## Figures and Tables

**Figure 1 nanomaterials-14-00372-f001:**
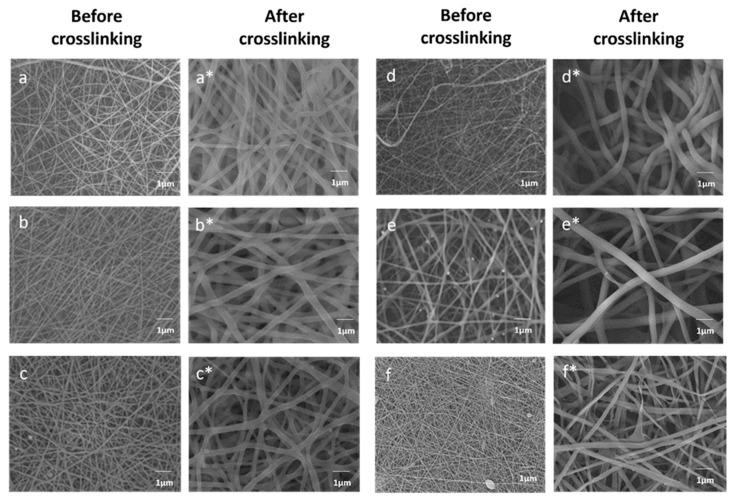
SEM micrographs of PVA scaffolds with different Bg concentrations, before and after (*) chemical crosslinking: (**a**,**a***) 5% hybrid, (**b**,**b***) 10% hybrid, (**c**,**c***) 15% hybrid; (**d**,**d***) 20% hybrid; (**e**,**e***) 25% hybrid and (**f**,**f***) 30% hybrid.

**Figure 2 nanomaterials-14-00372-f002:**
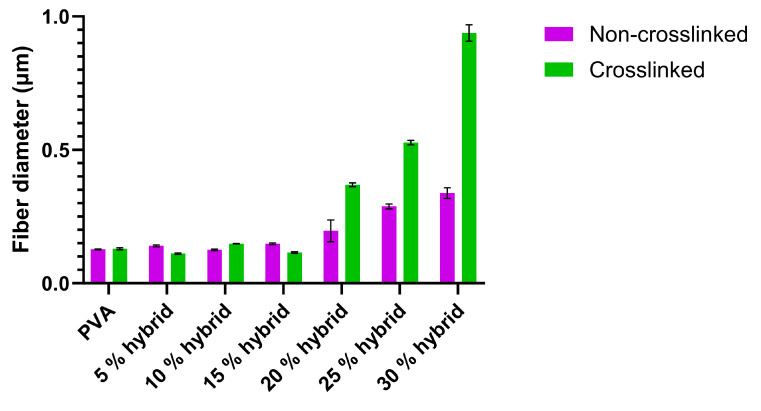
Shows the mean fiber diameters as a function of Bg concentration before and after chemical crosslinking with GA, measured with ImageJ software.

**Figure 3 nanomaterials-14-00372-f003:**
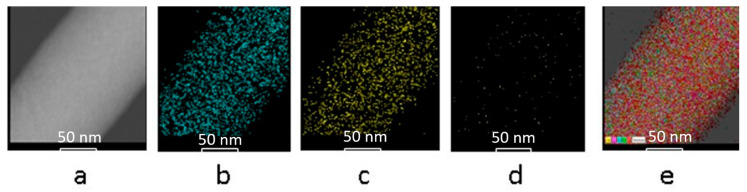
Elemental mapping in hybrids scaffold at 20% Bg concentration: (**a**) electron micrograph of PVA nanofiber, (**b**) Si (cyan), (**c**) Ca (yellow), and (**d**) P (green), and (**e**) merge image overlaying all elemental distributions on the nanofiber micrograph. Specific colors were added afterward.

**Figure 4 nanomaterials-14-00372-f004:**
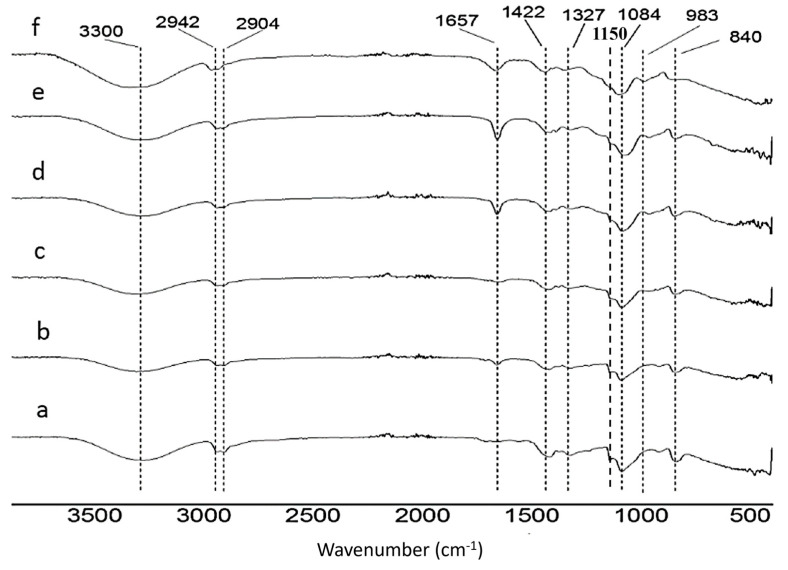
FTIR spectra of PVA scaffolds with different Bg concentrations: (**a**) 5% hybrid; (**b**) 10% hybrid; (**c**) 15% hybrid; (**d**) 20% hybrid; (**e**) 25% hybrid; and (**f**) 30% hybrid.

**Figure 5 nanomaterials-14-00372-f005:**
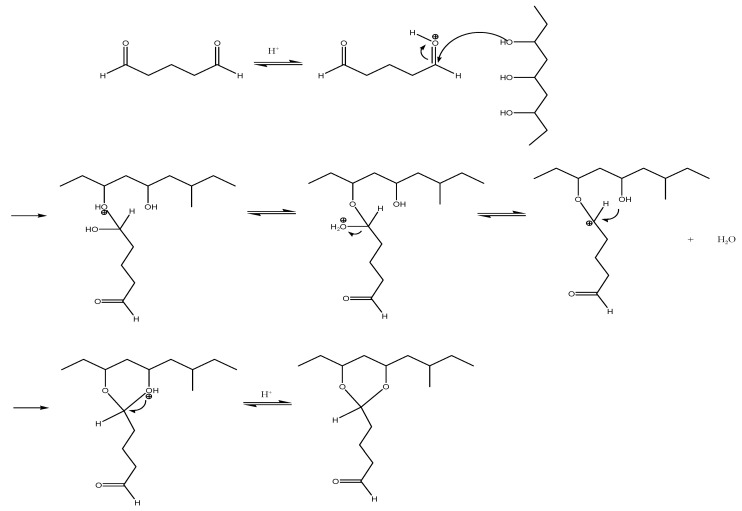
Schematic representation of the crosslinking reaction of PVA with GA.

**Figure 6 nanomaterials-14-00372-f006:**
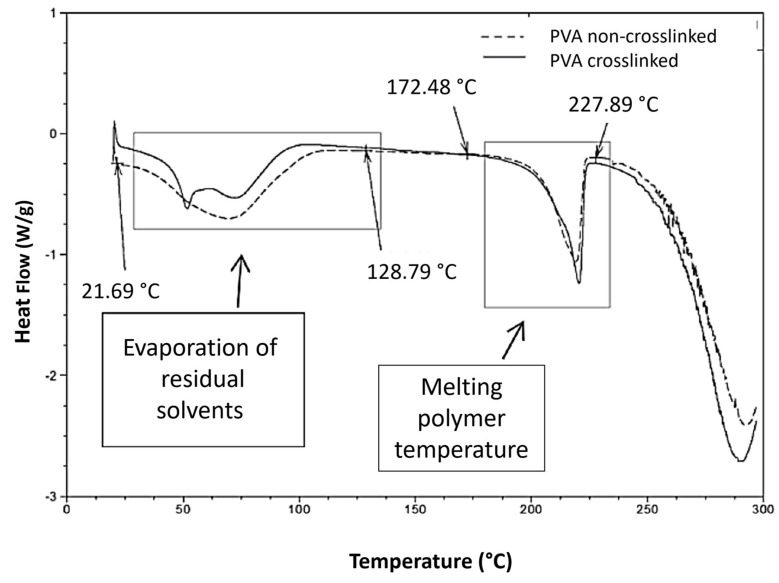
PVA DSC thermograms before and after crosslinking.

**Figure 7 nanomaterials-14-00372-f007:**
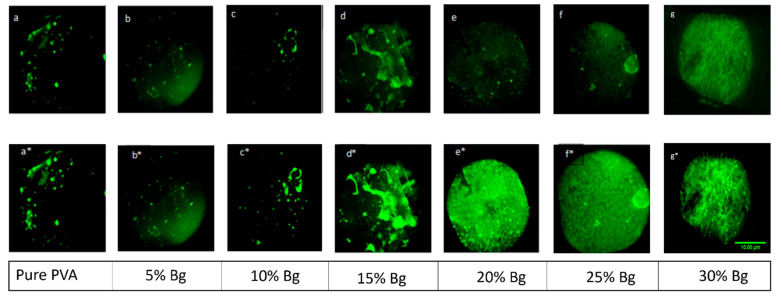
Fluorescence images of cardiomyocytes on hybrid PVA/Bg scaffolds, at different Bg concentrations: (**a**–**g**) at rest (top row) and (**a***–**g***) during contractile activity (bottom row). (4×) stereoscopic microscope.

**Figure 8 nanomaterials-14-00372-f008:**
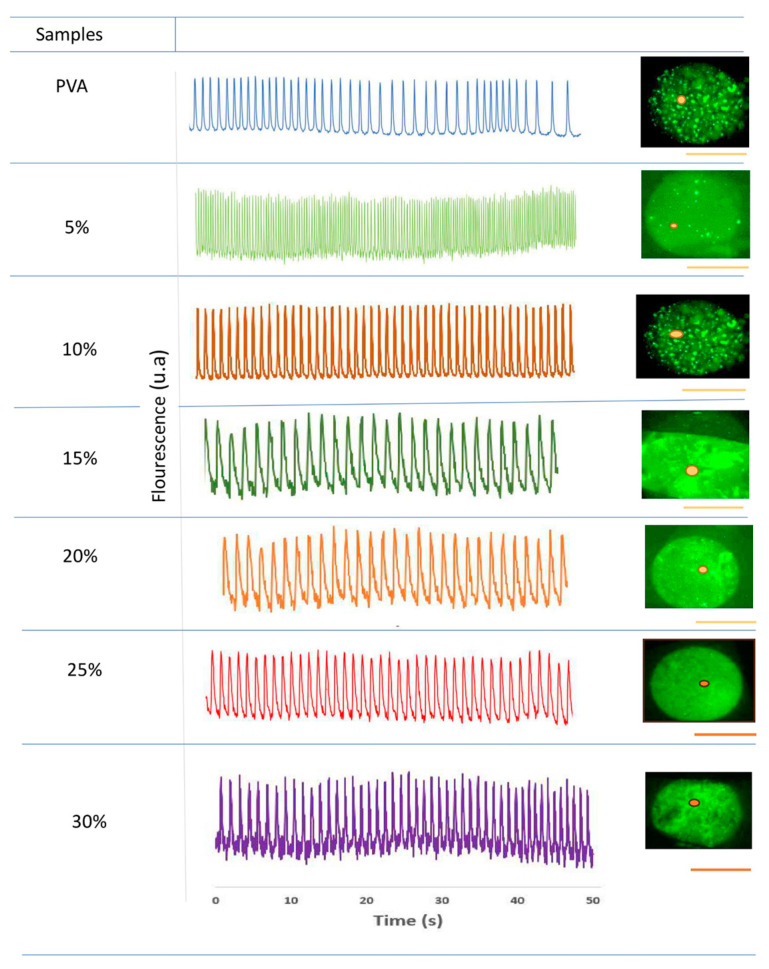
Contractile activity patterns revealed by fluorescence peaks attributable to intracellular ΔCa^2+^ are shown in the center panel; they were selected as regions of interest (ROIs) and are marked with a red dot on the corresponding image on the right. Right column: Fluorescence images of embryonic ventricular cardiomyocytes seeded on PVA/Bg-crosslinked scaffolds at six Bg concentrations as indicated in the left column. As can be appreciated, the different substrates do not inhibit contractile activity. However, the different Bg concentrations regulate adhesion and form separate cell aggregates (5, 10, and 15% Bg) or confluent layers (20, 25, and 30% Bg) with synchronized contractile activity in extended regions. Scale 100 μm.

**Figure 9 nanomaterials-14-00372-f009:**
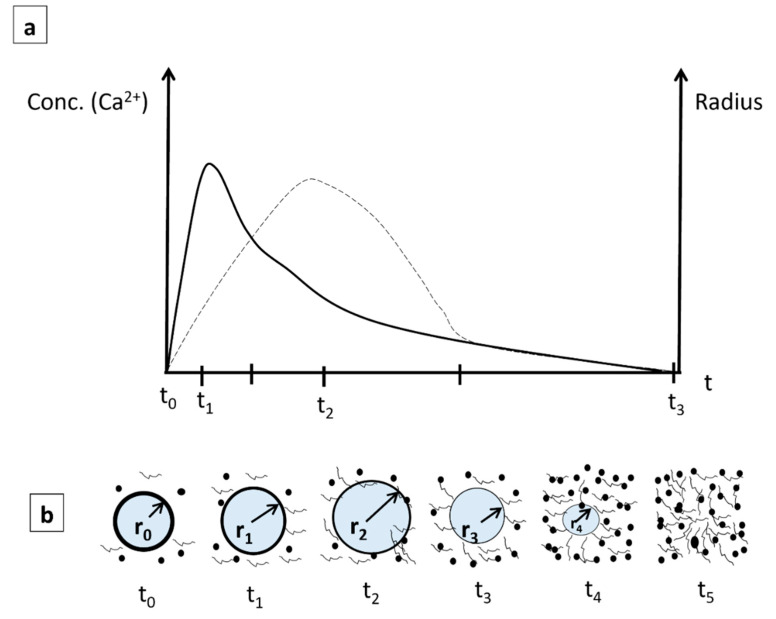
(**a**) Diagrams of dissolution of Ca^2+^ ions and fiber radius versus time: dissolution of Ca^2+^, solid line, and fiber radius (r), dotted line. (**b**) Schematic of the change in fiber radius at different times.

**Figure 10 nanomaterials-14-00372-f010:**
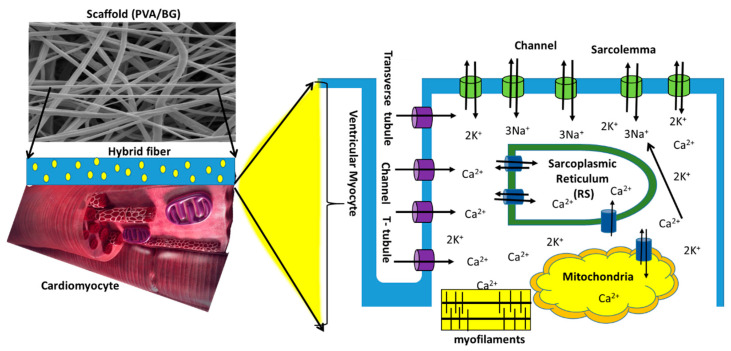
Schematization of cardiomyocytes across the PVA/Bg scaffold surface; cardiomyocytes adhere to the surface of nanofibers in the presence of a culture medium. Released Ca^2+^, Si^2+^, and PO_4_^3−^ surrounding the cardiomyocyte surface and other ions from the culture medium.

**Table 1 nanomaterials-14-00372-t001:** Composition of PVA/Bg solutions and calculated calcium concentration.

Hybrid Scaffold Name	PVA/Bg Ratio (*w*/*w*)	CaO Concentration in Bg	Ca^2+^ Concentration in Bg
PVA	100/0	0	0
5%	95/5	0.1590	0.0114
10%	90/10	0.3180	0.0227
15%	85/15	0.0477	0.0341
20%	80/20	0.0635	0.0454
25%	75/25	0.0794	0.0567
30%	70/30	0.0953	0.0681

**Table 2 nanomaterials-14-00372-t002:** Thermal transitions of PVA/Bg hybrids scaffolds.

Bg Content in PVA, (%)	Non-Crosslinked PVA			Crosslinked PVA	
	T_Ev_ (°C)	T_g_ (°C)	T_m_ (°C)	T_g_ (°C)	T_m_ (°C)
**Pristine PVA**	50	63	218	66	218
**(a) 5%**	-	63	219	66	219
**(b) 10%**	-	70	205	68	262
**(c) 15%**	-	73	260	69	300
**(d) 20%**	60	78	263	78	300
**(e) 25%**	61	88	254	92	300
**(f) 30%**	56	86	261	98	300
**Bioglass**	-	385	-	385	-

**Table 3 nanomaterials-14-00372-t003:** Weight loss determined by TGA of PVA/Bg hybrid scaffolds before (left) and after (right) crosslinking.

Bg Concentration in PVA, (%)	%Weight Loss before Crosslinking			%Weight Loss after Crosslinking		
	Temperature Intervals (°C)			Temperature Intervals (°C)		
	40–200	200–350	350–600	40–200	200–350	350–600
**5**	5	66	25	4.3	63.8	21.6
**10**	10	50	40	3	45	30
**15**	5	55	18	13	60	25
**20**	3	45	10	16	40	20
**25**	3	50	10	6	48	18
**30**	10	50	10	16	28	15
**PVA**	6	70	24	10	48	42
**Bg**	13.56	11	-	13.56	11	-

## Data Availability

Data are contained within the article and Appendix A.

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
