# Peer review of "In Vitro Modulation of Spontaneous Activity in Embryonic Cardiomyocytes Cultured on Poly(vinyl alcohol)/Bioglass Type 58S Electrospun Scaffolds"

_nanomaterials, 2024, doi:10.3390/nano14040372_

Round 1

Reviewer 1 Report

Comments and Suggestions for Authors

This paper is interesting, well-argued and fits well with the scopes of the journal.

By focusing on the realization of nanofibers based on poly (vinyl alcohol)/bioglass type 58S, it analyses the influences of filler amount on fibers morphology, thermal properties and biological interactions.

Data are clearly presented, and conclusions are consistent with both premises and discussion.

However, I have major revisions:

-        Pg 3 line 133: is the bio-glass sol gel synthesis a procedure already optimized in literature or for this specific activity?

-        Pg 4 line 167: what does “after the chemical reaction” means? How do you define that the chemical reaction is completed?

-        Pg 4 line 179: where are reported the hystograms

-        Pg 4 line 192: report in a more accurate way the cycle run performed for DSC and TGA analysis (i.e.: first run from #°C up to #°C at #°C/min, second run from #°C up to #°C at #°C/min, third run from #°C up to #°C at #°C/min)

-        Pg 5: in the abstract it is reported “this work aimed to modify the electrical conductivity in organic/inorganic electrospun hybrid scaffolds” therefore it could be interesting to show characterizations concerning the electrical conductivity of scaffolds.

-        Pg 6 line 249: high-resolution micrographs could be helpful to observe Bg nanoparticles on fibers surface.

-        Pg 6 and 7 Table 2: how do you explain the lowest diameter for PVA/Bg 10% and the decreasing values for PVA/Bg 5 and 15% after crosslinking?

-        Pg 7 line 287: report the FTIR-ATR spectra

-        Pg 9 line 309: report the DSC curves

-        Pg 10 line 332: report the TGA curves

-        Pg 14 Figure 6: is this graph deduced from literature or your experimental data?

-        The stability of your fibers in water should be analysed, in terms of diameter and mass variations, morphological and elemental analysis at different time interval.

-        Pg 15 line 451: cells adhesion on scaffold surface should be showed as well cell viability

Reviewer 2 Report

Comments and Suggestions for Authors

General comments: The article entitled "In vitro modulation of spontaneous activity in embryonic cardi-2 omyocytes cultured on poly (vinyl alcohol)/bioglass type 58S 3 electrospun scaffolds" The discusses research on the potential restoration of biological functions of cardiac tissue using hybrid nanofibers composed of poly (vinyl alcohol) (PVA) and bioglass type 58S. The study reveals that hybrid scaffolds with a 25% bioglass content improve thermal stability, reduce degradation in water, and enhance adhesion and contractility of cardiomyocytes, what is interesting result. Although the presented results, especially the biological part, are promising, the article requires supplementary data and discussion.

Major comments:

1.       The described mechanism of fiber swelling after the cross-linking process is unclear. Authors should expand on this paragraph with reference to the literature.

2.       Why do the authors change the weight fraction of PVA relative to Bg? Based on such compositions, it is difficult to discuss the effect of the addition of Bg if the weight fraction of the polymer in each system is different? Was the GA concentration for each system the same or was it converted into relative PVA?

3.       The authors describe changes in band intensity in FTIR spectra without presenting them (line 299 and subsequent lines). Due to this, the reader cannot see what is being said. Please include spectra in supplementary file.

4.       The crosslinking process results in the formation of C-O-O bonds – where are they represented on the spectra?

5.       The discussion of thermal properties (DSC) relates to the degree of hydrolysis, while the main factors influencing these properties will be the amount of bioglass added and the degree of cross-linking. The degree of hydrolysis is related to the hydrolysis (or alcoholysis) of poly(vinyl acetate) (PVAc) and the "consumption" of OH groups for the crosslinking process does not change the degree of hydrolysis.

6.       Similarly, to the description of FTIR, also in the description of TGA results the authors refer to thermograms which are not presented, please add the thermograms to the supplementary file with proper reference in the text. Why is the mass loss after cross-linking 103% for the PVA + 5% BG system?

Minor comments:

1.       Please add information how the samples were prepared for SEM and TEM observations – where coated or not, how the fibers were placed on the TEM grid?

Comments on the Quality of English Language

English is correct.

Reviewer 3 Report

Comments and Suggestions for Authors

Please find my comments and doubts in the attachment.

Round 2

Reviewer 1 Report

Comments and Suggestions for Authors

I appreciate the reponses and the modifications in the main text and in the SI file. However I suggest to add in the text the explanation reported in Response 2 and 5. As concerning comment 6 it could be interesting also to report the particles size distribution in order to explain why it is not possible to detect bioglass particles by SEM analysis. As regard comment 10 and Figure S6 I cannot understand the graphs, why the starting point is not 100?

Lastly, I consider useful and fundamental to show in the main text that fibers are able to withstand the biological environment and for the time required to biological response.

Reviewer 2 Report

Comments and Suggestions for Authors

I have no comments after the authors' additions.

Reviewer 3 Report

Comments and Suggestions for Authors

I couldn't find the answer to the question about fluorescence imaging of cardiomyocytes on hybrid PVA/Bg scaffolds. Polymer scaffolds tend to absorb fluorescent dyes. Are the authors sure about the interpretation of the results of contractile activity? 

Still parameters i.e. voltage, current are missing in EM method section. 
